# Cost Analysis of Prefabricated Elements of the Ordinary and Lightweight Concrete Walls in Residential Construction

**DOI:** 10.3390/ma12213629

**Published:** 2019-11-04

**Authors:** Marzena Kurpinska, Beata Grzyl, Adam Kristowski

**Affiliations:** Faculty of Civil and Environmental Engineering, Gdańsk University of Technology, Narutowicza 11/12 Str., 80-233 Gdańsk, Poland; marzena.kurpinska@pg.edu.pl (M.K.); beata.grzyl@pg.edu.pl (B.G.)

**Keywords:** building, prefabricated concrete wall, lightweight concrete, recycled aggregates, economic analysis

## Abstract

Global economic growth causes an increase in natural resources exploitation, particularly in construction branch. The growing use of electricity contributes to climate change. Therefore, it is necessary to search the solutions, which will allow for reducing natural resources exploitation. One of the many opportunities to do that is the application of the recycled materials. The authors of the given article have analyzed three variants of construction solutions. One of them was the production of the walls of a building from reinforced concrete prefabricates with styrofoam insulation layer. The second variant for analysis were prefabricated walls from lightweight concrete, made of sintered clay aggregate with a foam core. The third proposed variant was a system of multi-layered walls, which was made of lightweight concrete with granulated expanded glass aggregate (GEGA). The main objective of the research was to assess the use of lightweight GEGA prefabricates, focusing on economic and technological aspects of the solution. The authors have analyzed the entire construction costs; ceilings and stairs were assumed as reinforced concrete elements. In calculations, the weight of the elements was taken into account, as well as transportation and mounting costs. On the basis of this cost analysis, it was concluded that the use of prefabricated element, made of lightweight concrete with GEGA, could be a replacement for the solutions, widely applied until these days. The analysis has also shown that the use of prefabricates with GEGA is sensible from the economic viewpoint, as it allows for saving construction time. Moreover, the solutions, proposed here, allow for saving natural resources and assuming a more environmentally friendly and caring attitude.

## 1. Introduction

Science and technology development enable the implementation of innovative solutions in construction and architecture. Multiple changes in concrete production and prefabrication are being introduced. At present, concrete as a modern composite is regarded not only as a construction material, but also as an insulation material or aesthetic architectural finish. Concrete properties depend on concrete mixture components.

Concrete technologists face increasing expectations, as more and more modifications of both ready-mix and prefabricates become possible to be made, maintaining material’s durability and strength at the same time. It is also possible to modify cement binder by means of applying admixtures and additives or natural, artificial, or recycled aggregates alongside with sustainable development principles while obtaining components, producing ready-mix and prefabricating elements.

New technologies in artificial lightweight aggregate production enable meeting complex requirements of the construction contractors as for strength, insulation, acoustic, fire-resistance, and aesthetic properties. According to [1], lightweight concrete is a concrete with not less than 800 kg/m^3^ and not more than 2000 kg/m^3^ volume density in a dry state and with strength from LC 8/9 to LC 80/88. High strength lightweight concrete is a concrete with compression strength that is higher than LC50/55. Currently, due to the possibility of the use of the additives and chemical admixtures, it is possible to obtain considerably higher strength parameters for lightweight concrete than is required by the standard. Lightweight concrete and its modifications are subject to a great number of research at the moment [2,3,4,5,6,7]. Due to the possibility of application of the prefabricated elements, made of lightweight concrete as construction elements, it is necessary to design them with a great deal of precision and take into account the requirements that specifically apply to the type of construction. The components of a mix should be selected with the thought of safety of the future users of the construction and, also, the durability of the elements and their physical and mechanical properties. Basic features of lightweight concrete, which make a difference, are lower volume density and better insulation properties [8,9,10,11]. Lightweight concrete with its volume density of about 1800 kg/m^3^ shows thermal insulation properties that are similar to those of ceramic brick wall [12,13]. The use of lightweight concrete in prefabricates results in the building’s specific weight reduction and thermal and acoustic insulation increase of a building [14,15,16]. The application of a lightweight aggregate, e.g., foam glass aggregate, allows for a significant decrease in concrete volume density and thermal insulation coefficient reduction [17,18,19,20,21].

## 2. Materials and Methods

### Prefabricated Concrete and Residential Buildings

Prefabrication gives great opportunities for public, residential, and industrial construction. The essence of this technology lies in the optimization of construction process, safety procedures improvement, limitation of the waste production, and minimization of exhaust gas emission. Moreover, the search of the effective methods of construction project realization in keeping with the aesthetic and comfort values of the constructed object is of high importance. The aim of prefabrications is saving construction time, reduction of the number of construction processes, minimization of construction’s cross section, and quality improvement [22,23,24,25]. Prefabricated elements are widely used in residential construction as multi-layered walls, ceilings, and balcony panels due to their numerous advantages. The prefabricate production process led, in modern conditions, strictly following technological procedures, going on non-dependent on atmospheric conditions production floors, allows for receiving perfect quality products. However, it is crucial to bear in mind the necessity of strict overall investment control and preliminary assessment of the difficulties while mounting. Economic analysis of the investment is also obligatory, so as to make sure of the financial reasonability of the project [26].

Table 1 presents a comparison of the building erection system by means of the method of performing all activities at the construction site and using prefabricated elements.

The prefabricated multi-layer wall is frequently used in the prefabricated integrated residential buildings, which are characterized by easy installation and short construction time [27,28,29]. Figure 1 shows the process of manufacturing a prefabricated wall element.

Further analyzing the case of installing prefabricated walls in an apartment building, thermal conductivity aspects were considered. Heat transfer coefficient for the walls was determined, while taking into account higher requirements, connected with thermal insulation in the countries with occurring problem of significant indoor and outdoor temperature difference. Table 2 shows the requirements for meeting for thermal conductivity of outer and inner walls of the residential building.

According to the regulations, connected with thermal insulation of the buildings, the maximum total heat transfer coefficient of the walls between the corridors and apartments constitutes U = 1.0 W/(m^2^·K). In the case of outer walls, depending of the assumed temperature for calculations, the coefficient U = 0.3 W/(m^2^·K) or U = 0.8 W/(m^2^·K). These requirements cannot be met by one-layer reinforced-concrete wall or brick wall. One could apply a thermal insulation layer of foam panels or mineral wool or use thermal insulation plaster in order to meet the latter thermal insulation requirements. However, these solutions will generate additional costs and will make the walls thicker. Additional construction works, connected with thermal insulation, will cause construction deadlines prolongation. Therefore, an alternative solution lies in using multilayer prefabricated elements, which are made of lightweight concrete.

Table 3 presents the values for calculation of the physical properties of the selected materials, determined according to [31] and this article’s authors’ own research.

Figure 2 presents the types of prefabricated walls.

## 3. Results

### 3.1. Comparison of the Heat Transfer Coefficient through Outer Walls

The types of layers, thickness of the layers, the values of thermal conductivity coefficients, and thermal resistivity values were determined to compare and analyze the values of the heat transfer coefficients for the three types of a vertical outer wall. The walls with the ordinary concrete post and the post, made of concrete with expanded clay aggregate were compared, as well as the ones made of concrete with fly ash aggregate and with granulated foam glass aggregate.

### 3.2. Thermal Insulation

The total thermal resistivity *R_T_* of the flat construction component, consisting of homogeneous insulation layers, perpendicular to the direction of the heat flow should be calculated according to Equation (1).
*R*_1_ = d_1_/λ_1_; *R*_2_ = d_2_/λ_2_; … *R_i_* = d_i_/λ_i_(1)
where:
d_1_, d_2_, … d_i_—element thickness (cm)λ_1_, λ_2_, … λ_i_—thermal conductivity coefficient (W/m·K)
*R_T_* = *R_si_* + *R*_1_ + *R*_2_ +…..+ *R_i_* + *R_se_*(2)
where:
*R_T_*—total thermal resistivity (m^2^·K/W)*R**_si_*—thermal resistance on the inner surface, (m^2^·K/W)*R*_1_, *R*_2_, … *R_i_*—each layer’s thermal resistivity values, (m^2^·K/W)*R_se_*—thermal resistance on the outer surface, (m^2^·K/W)

Heat transfer coefficient was calculated according to the Equation (3).
(3)$$U=1/R_{T}\;(W/m^{2}\cdot K)$$

The values of thermal resistance *R_si_*, *R_se_* [32,33] were assumed according to Table 4.

Heat transfer coefficient of the wall was calculated according to the Equation (4).
(4)$$U=\frac{1}{R_{si}+R_{1}+R_{2}+R_{3}+\ldots R_{i}+R_{se}}$$

The main assumption was the maximum heat transfer coefficient U_max_ could not be higher than 0.2 (W/(m^2^·K)).

## 4. Discussion

The research of different variants of the project realization and the proposal of the optimal one leads to the choice of the best solution from the point of view of the chosen criterion. The aim of the alternative solution analysis is, among others, to answer the question, if the new technology or material implementation is beneficial from the investor’s/manufacturer’s point of view and if it leads to the cost reduction and/or production time saving [34,35]. In the construction branch, cost reduction and/or construction elements production time saving is most frequently obtained through the use of the relevant materials and through mounting process optimization, and, as a result, the overall time saving [36].

In order to optimize costs, it is advised to prepare construction cost estimation, including all costs calculation (labor, materials, equipment, indirect costs, and profit) for the solutions in question. In the case of time optimization, it is necessary to analyze workload for each process within the proposed solutions.

The objective of the analysis, presented in the following part of the article, is to compare the labor cost and time needed for mounting the elements of the construction. Three solutions have been proposed, for which the costs and time of realization were analyzed.

### 4.1. The Cost Analysis of the Facility Implementation in Prefabricated Technology

The object was selected, the bill of quantities was prepared, assumptions for calculation were made, and a cost-estimate calculation was carried out in order to determine the cost of raising a building in a raw state by means of prefabricated technology and the use of three types of materials. Three solutions were analyzed.

The aforementioned methods were applied on an actual construction site. Subsequently, the economic, environmental, and social benefits of prefabricated concrete application in residential construction were analyzed.

Solution 1 consists of assembling the object from prefabricated reinforced concrete elements, which are traditionally reinforced with two steel meshes from main bars ϕ10 mm every 10 cm (vertical reinforcement) and from distribution bars ϕ8 mm every 15 cm (horizontal reinforcement). A sandwich wall contains an insulating layer of polystyrene. In addition, the wall has elements for transporting and arranging elements. Table 5 provides detailed information.

Solution 2 is an assembly of the object, using prefabricated layered elements, with a lightweight concrete core of sintered and expanded clay (Leca). Wall elements are reinforced with two steel meshes from main bars ϕ10 mm every 15 cm (vertical reinforcement) and distribution bars ϕ6 mm, laid every 15 cm (horizontal reinforcement). In addition, the wall has transporting and arranging elements. Table 5 provides detailed information.

Solution 3 consists of assembling the object with sandwich elements with a lightweight concrete core from granulated expanded glass aggregate, with an insulating layer of ultra-light concrete containing perlite and granulated expanded glass aggregate from the outside. Gypsum plaster was used in the inside. The wall reinforcement was designed with the 8 mm main bars every 15 cm (vertical reinforcement) and with ϕ6 mm distribution bars being laid every 15 cm (horizontal reinforcement). In addition, lightweight reinforced polymer structural fiber reinforcement was used in the amount of 2 kg/m^3^. The wall has elements, which enable the transport and arrangement of elements. Table 5 provides detailed information.

In Table 5, one can see the examples of the walls alongside with heat transfer coefficient.

As an example, a multi-family, five-story, three-frame residential building was subjected to economic, technological, and organizational analysis (Figure 3). There are 30 residential premises with a total usable floor area of 1664.90 m^2^. The built-up area is 520.80 m^2^, the building has a cubature of 8900.60 m^2^, and the total area is 2983.40 m^2^ (including 76.20 m^2^ of service and technical rooms). The building is constructed of prefabricated elements in large-panel technology. The structural arrangement of the load-bearing walls is transverse, their spacing is 3.0 and 4.8 m. On the last storey of the building, masonry, prefabricated, and wet cast elements were used. Foundation benches are of reinforced concrete, wet cast from C30/37, concrete, and exposure class XC2. Structural walls of the underground and above-ground storeys are made of 15 cm thick prefabricated reinforced concrete elements.

On the basis of the project documentation, a bill of quantities (BOQ) was prepared, the range and quantity of certain works, needed for the raw object state, were determined. On that basis, an investor’s cost-estimate calculation was done in a simplified version. In both BOQ and estimation the production cost of the walls, ceilings and stairs were calculated (Table 6). The assessment was made on the basis of [37] and individual calculations [38]. It was assumed that, given retail prices would include labor, material, equipment costs, as well as indirect costs and profit for a mounting unit of each element [39]. For the walls, the retail prices concerned three proposed solutions (1–3). It was assumed that the production of the ceilings and stairs would be, in each case, as for solution 1 (prefabricated reinforced concrete).

Price per unit: Solution 1Prefabricated reinforced concrete (walls, ceilings and stairs): 486.42 monetary unitsSolution 2Prefabricated items from concrete products made of sintered and expanded clay aggregate (walls) and prefabricated reinforced concrete (ceilings and stairs): 494.75 monetary unitsSolution 3Prefabricated products that were made of lightweight concrete with GEGA (walls) and prefabricated reinforced concrete (ceilings and stairs): 512.87 monetary units.

Based on the cost calculation, prepared by means of the simplified method and prices from the third quarter of 2019 from the Sekocenbud price list [39] and producer prices, the cost of implementing the basic elements of the building’s raw state was made according to the three proposed solutions.

The cost of making the above range of the building shell (walls, ceilings, stairs) from reinforced concrete precast elements is 998,855.19 (monetary units). The cost of the same scope of works from precast lightweight concrete, which was made of sintered and expanded clay, is 1,039,454.82 (monetary units). The cost of prefabricated lightweight concrete works is 1,128,065.67 (monetary units). Solution No. 2 is 4.1% more expensive than solution No. 1, while solution No. 3 is more expensive than solution No. 1 by 12.9%. (Table 6, Figure 4).

### 4.2. The Analysis of Assembly Time

Table 7 presents a summary of the project’s implementation times according to the three proposed solutions. The following assumption was made: in each case, the construction of five floors of the building, five to 14 employees are employed (depending on the scope and nature of technological processes).

The expenditure of time given in Table 6 was determined on the basis of Material Expenditure Catalog KNR 2-02 [40] and own analyzes [38,41].

Based on the information contained in Table 7 and Figure 5, it can be concluded that the expenditure of working time for the technology of assembling the object according to solutions 2 and 3 are similar and significantly differ from the expenditure of time for solution 1. This is due to the fact that the structural elements—prefabricated concrete lightweight, made of sintered and expanded clay and lightweight concrete precast elements, made of GEGA, have similar weight and they are definitely lighter than reinforced concrete precasts. The assembly time of the object according to solution 3 is 4.3% shorter than the time resulting from solution 1.

## 5. Conclusions

The analysis, carried allows for formulating the following conclusions.

1. In practice, it is possible to use a prefabricated building, made of lightweight concrete, while maintaining all of the requirements for the construction of a building. This solution, in a broader context, finds technical and economic justification for its implementation in practice. It is possible to achieve time savings of up to 5% when compared to the assembly of a reinforced concrete precast object (solution 1).

2. The main advantage of using broadly defined prefabricated technology (compared to monolithic technology) is the significant time saving that results from the incorporation of ready-made construction products. The analysis, as presented in the article for the three solutions, involving the facility assembly from ready prefabricated elements, showed that, depending on the type of applied material, of which elements are made, it is possible to achieve a shorter implementation time of a given scope of construction works. With large-scale works, it is possible to achieve significant savings in investment implementation time. This applies to the prefabricated transport process to the construction site. It is possible to transport a larger number of them by the transport unit at one time due to the lower weight of the elements, made of lightweight concrete, which in turn also speeds up the delivery and assembly time on site, reduces fuel consumption, and achieves lower material transport costs. Solution 3, therefore, reduces investor’s expenses in this area and significantly increases the pace of works at the construction site. In practice, the choice of solution 1 from three presented ones is associated with the lowest cost for the investor, but the longest time of work performance—embedding prefabricated elements.

3. Solution 3 includes energy saving and the efficient use of raw materials in the production and assembly of prefabricated elements. It fully fits in with the idea of environmental protection and sustainable development of the economy [42,43]. Its use allows for reducing CO_2_ emissions in the process of manufacturing prefabricated elements and their transport to the construction site, and thus reduces environmental degradation [44]. The implementation of reinforced concrete precast elements (solution 1) requires greater energy consumption, which is necessary for producing a large amount of concrete mix. By using lightweight concrete, it is possible to use raw materials from the recycling of glass or fly ash. This significantly reduces environmental degradation [45,46], including large-scale mining of the aggregates.

4. The use of technology using solutions 2 and 3 affects the highest quality of performed work and ensures optimal thermal conductivity and fire resistance.

5. In construction practice, the integrated assembly of prefabricated elements, made of lightweight concrete, has not yet gained great interest. The numerous, also long-term benefits of using solution 3 should, however, encourage investors to choose and make extensive use of precast lightweight concrete, despite their higher price.

## Figures and Tables

**Figure 1 materials-12-03629-f001:**
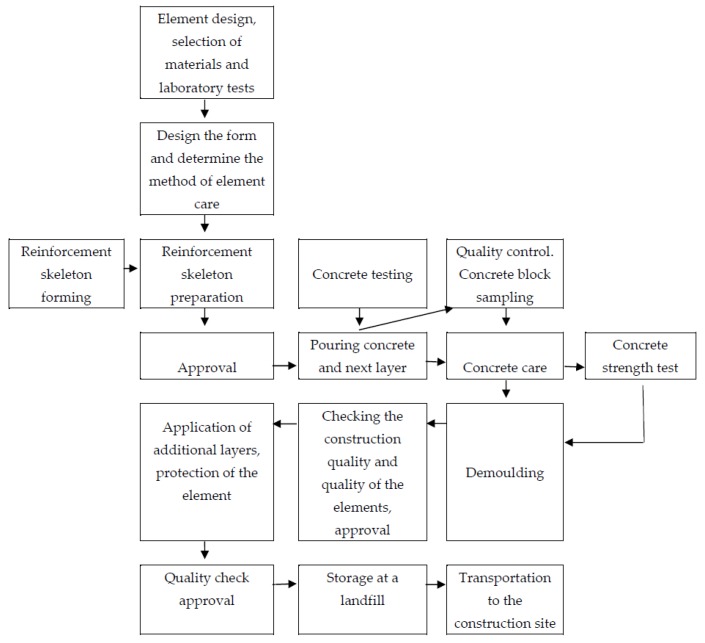
Process of manufacturing of prefabricated wall elements [30].

**Figure 2 materials-12-03629-f002:**
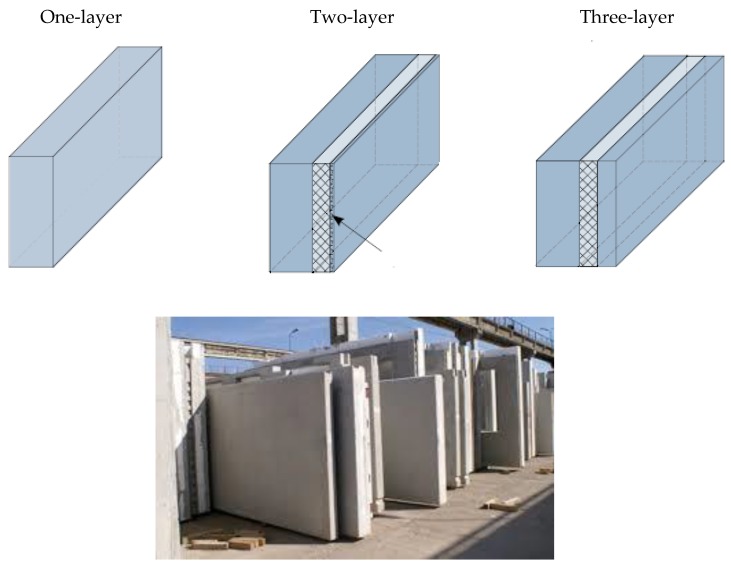
Prefabricated elements storage.

**Figure 3 materials-12-03629-f003:**
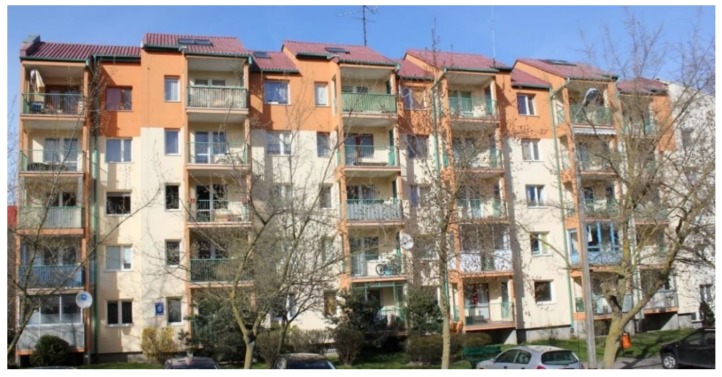
A multi-family, five-story, three-frame residential building. Source: own photo of the authors.

**Figure 4 materials-12-03629-f004:**
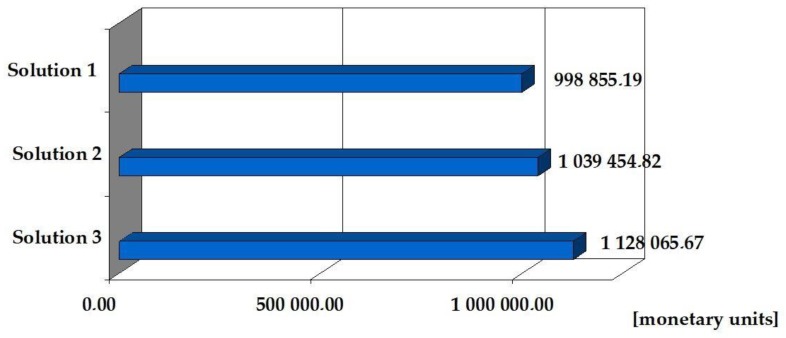
Cost of making walls, ceilings and stairs in the building implemented according to three solutions.

**Figure 5 materials-12-03629-f005:**
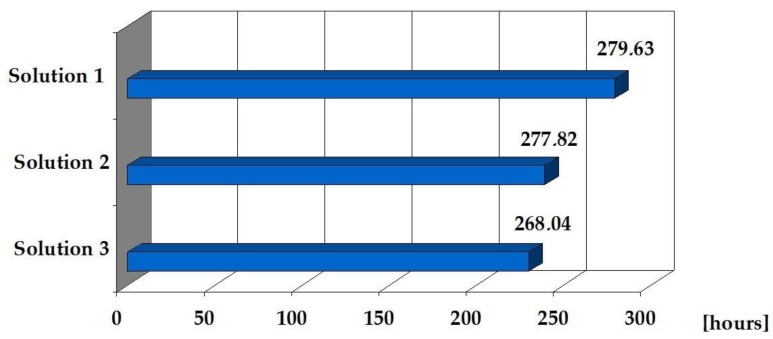
Time of processing walls, ceilings, and stairs in the building to be implemented according to three solutions.

**Table 1 materials-12-03629-t001:** Comparison between on-site and off-site construction based on [26].

	On-Site Construction	Off-Site Construction
Labor/time	Labor work intensive. Longer time for construction.	Technology intensive. Shorter time for construction.
Environmental independence	Remarkably influenced by the ambient temperature and other factors.	Prefabricated components can be directly assembled on site.
Quality control	Hard to find an agreed standard for various situations.	Quality can be easily controlled, the elements are repeatable.
Shape flexibility	On-site construction is often applied for buildings with complicated designs.	Buildings are relatively alike due to fixed scale.
Construction management	Complex management of material stocking, human resources and safety.	Transport of the materials and their stocking can be reduced.
Resource consumption	Low efficiency of resource usage. Huge energy consumption.	Industrialization of components increases the efficiency of resource usage. A specified factory is usually needed.
Environmental friendliness	Noise and pollution influence the environment greatly.	Rare noise and pollution, hence, more environmentally friendly.
Construction function	Special procedures need to be applied for water and fire protection. Lower construction efficiency.	Components of specified functions are precast in factory, which reduces difficulties.
Structure performance	Better performance in integrality and stability.	Relatively weaker in stability and earthquake-resistance if we use ordinary concrete and better if we use lightweight concrete.

**Table 2 materials-12-03629-t002:** The values of the heat transfer coefficient U (W/m^2^·K) for the walls of the residential buildings, determined for weather conditions in Poland (according to Ministerial Order §134 art. 2).

Kind of the Wall and Indoor Temperature	Heat Transfer Coefficient U_max_ (W/m^2^·K)
Outer walls (exposed to the outdoor air) (a) at t_i_ >16 °C (b) at t_i_ ≤ 16 °C where: t_i_—indoor temperature	0.3 0.8
Inner walls between the heated and unheated rooms, staircases and corridors	1.0

**Table 3 materials-12-03629-t003:** Values for calculation of physical properties of the selected materials.

Material	Density in a Dry State ρ (kg/m^3^)	Thermal Conductivity Coefficient λ (W/m·K)
Ordinary concrete	2200	1.30
Ordinary concrete with steel rebar (2%)	2400	1.70
Concrete with expanded clay aggregate	1000	0.39
	1100	0.46
	1200	0.54
	1300	0.62
	1400	0.72
	1600	0.90
Concrete with granulaed foam glass aggregate and sand *	1000	0.39
Concrete with granulated foam glass aggregate and perlite with dispersed fiber reinforcement	600	0.38
	800	0.46
	1000	0.51
Concrete with granulaed foam glass aggregate and granilated sintered fly ash aggregate *	1000	0.54
	1200	0.60
	1400	0.67
	1600	0.74
Styrofoam (EPS)	12	0.045
	15	0.043
	20	0.040
Mineral wool	50	0.038
	90	0.039
	130	0.040
Cement-lime plaster	1850	0.90
Gypsum plaster	1000	0.40
	600	0.18
Air	1.23	0.025

* Authors’ own research.

**Table 4 materials-12-03629-t004:** Values of the thermal resistance *R_si_*, *R_se_*.

Thermal Resistance	Direction of the Heat Flow		
	Horizontal	Horizontal (up)	Vertical (down)
*R_si_* (m^2^·K/W)	0.13	0.10	0.17
*R_se_* (m^2^·K/W)		0.04	

**Table 5 materials-12-03629-t005:** Heat transfer coefficients for an outer multilayer wall.

Element Sort	Layer Sort	Material Density	Layer Thickness	Thermal Conductivity Coeficient for the Material λ	Layer Thermal Resistivity Ri	Heat Transfer Coefficient for the Wall U	Weight for Wall’s Dimentions (6 × 3 m^2^)
		(kg/m^3^)	(m)	(W/m^2^·K)	(W/m^2^·K)	(W/(m^2^·K))	(kg)
Ordinary concrete wall Class C20/25 (wall thickness 39 cm)	Rse	-	-	-	0.04	0.20	7524.4
	Cement-lime plaser	1850	0.015	0.82	0.02		
	Ordinary concrete with a rebar	2400	0.15	1.7	0.09		
	Styrofoam	12	0.21	0.045	4.67		
	Cement-lime plaser	1850	0.015	0.82	0.02		
	Rsi	-	-	-	0.13		
Lightweight concrete wall with expanded clay aggregate and natural sand LC 20/22 (wall thickness 43 cm)	Rse	-	-	-	0.04	0.20	6228.4
	Cement-lime plaser	1850	0.015	0.82	0.02		
	Concrete with expanded clay aggregate (1600)	1600	0.18	0.9	0.20		
	Styrofoam	12	0.21	0.045	4.67		
	Cement-lime plaser.	1850	0.015	0.82	0.02		
	Rsi	-	-	-	0.13		
Lightweight concrete wall with granulated foam glass aggregate LC 20/22 (wall thickness 41.5 cm)	Rse	-	-	-	0.04	0.20	4705.2
	Lightweight insulation concrete with granulated foam glass aggregate and perlite (800)	800	0.05	0.46	0.11		
	styrofoam	12	0.2	0.045	4.44		
	Lightweight concrete with granulated foam glass aggregate and fly ash aggregate (400)	1400	0.15	0.67	0.22		
	Gypsum plaster (600)	600	0.01	0.18	0.06		
	Rsi	-	-	-	0.13		

**Table 6 materials-12-03629-t006:** List of structural elements for five storeys of the building, unit prices and total cost of walls, ceilings and stairs according to solutions 1, 2, 3.

Type of Item	Number (m^2^) or (pcs) of Elements of a Given Type on a Typical Story	Number (m^2^) or (pcs) of Elements of a Given Type in the Building	Solution 1 Prefabricated Reinforced Concrete		Solution 2 Prefabricated Items from Concrete Products Made of Sintered and Expanded Clay Aggreate		Solution 3 Prefabricated Products Made of Lightweight Concrete with GEGA	
			Price per Unit (Monetary Units)	Total Cost (Monetary Units)	Price per Unit (Monetary Units)	Total Cost (Monetary Units)	Price per Unit (Monetary Units)	Total Cost (Monetary Units)
(1)	(2)	(3)	(4)	(5) = (3) × (4)	(6)	(7) = (3) × (6)	(8)	(9) = (3) × (8)
External load-bearing walls	57.50 (m^2^)	287.50 (m^2^)	247.49	71,153.38	264.22	75,963.25	300.74	86,462.75
Outer curtain walls	216.00 (m^2^)	1089.00 (m^2^)	226.01	246,124.89	241.29	262,764.81	274.64	299,082.96
Internal load-bearing walls	220.30 (m^2^)	1101.50 (m^2^)	233.57	257,277.36	249.36	274,670.04	283.82	312,627.73
Internal partition walls	33.63 (m^2^)	168.15 (m^2^)	154.58	25,992.63	165.03	27,749.79	187.84	31,585.30
Ceiling with wreaths	416.80 (m^2^)	2084.00 (m^2^)	159.52	332,439.68	159.52	332,439.68	159.52	332,439.68
Stairs—running boards	6 (pcs)	30 (pcs)	2007.35	60,220.50	2007.35	60,220.50	2007.35	60,220.50
Stairs—landing plates	3 (pcs)	15 (pcs)	376.45	5646.75	376.45	5646.75	376.45	5646.75
			**Sum: 998,855.19** **(monetary units)**		**Sum: 1,039,454.82 (monetary units)**		**Sum: 1,128,065.67 (monetary units)**	

**Table 7 materials-12-03629-t007:** Time analysis—assembly of prefabricated elements according to three solutions.

Type of item	Number (m^2^) or (pcs) of Elements of a Given Type on a Typical Story	Number (m^2^) or (pcs) of Elements of a Given Type in the Building	Solution 1 Prefabricated Reinforced Concrete			Solution 2 Prefabricated Items from Concrete Products Made of Sintered and Expanded Clay Aggregate			Solution 3 Prefabricated Products Made of Lightweight Concrete with GEGA		
			Working Time per Work Unit [r-g]*	Working Time per Work Unit [m-g]*	Total Assembly Time (h)	Working Time per Work Unit [r-g]*	Working Time per Work unit [m-g]*	Total Assembly Time (h)	Working Time per Work Unit [r-g]*	Working Time per Work Unit [r-g]*	Total Assembly Time (h)
(1)	(2)	(3)	(4)	(5)	(6)	(7)	(8)	(9)	(10)	(11)	(12)
External load-bearing walls	57.50 (m^2^)	287.50 (m^2^)	0.172	0.051	14.21	0.170	0.050	14.00	0.157	0.042	12.21
Outer curtain walls	216.00 (m^2^)	1089.00 (m^2^)	0.165	0.042	45.94	0.153	0.040	45.20	0.148	0.036	43.81
Internal load-bearing walls	220.30 (m^2^)	1101.50 (m^2^)	0.165	0.042	44.95	0.165	0.040	44.15	0.155	0.036	42.82
Internal partition walls	33.63 (m^2^)	168.15 (m^2^)	0.143	0.041	7.11	0.140	0.040	7.05	0.125	0.037	6.79
Ceiling with wreaths	416.80 (m^2^)	2084.00 (m^2^)	1.180	0.063	150.20	1.180	0.063	150.20	1.180	0.063	150.20
Stairs—running boards	6 (pcs)	30 (pcs)	1.053	0.374	12.21	1.053	0.374	12.21	1.053	0.374	12.21
Stairs—landing plates	3 (pcs)	15 (pcs)	0.752	0.330	5.01	0.752	0.330	5.01	0.752	0.330	5.01
**Sum: 279.63 (h)**						**Sum: 277.82 (h)**			**Sum: 268.04 (h)**		

[r-g]*—man hours; [m-g]*—machine hours.

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
