# Peer review of "Cost Analysis of Prefabricated Elements of the Ordinary and Lightweight Concrete Walls in Residential Construction"

_materials, 2019, doi:10.3390/ma12213629_

Round 1

Reviewer 1 Report

line 21,” Granulated Foam Glass Aggregate”,suggest change to ” granulated foam glass aggregate” What is ”??” in Table 3與equation (2)? 4.1. The cost analysis of the facility implementation in prefabricated technology. Can author provide sketch for suggested solutions including dimension, reinforcing bars and concrete materials). Line 180-182, ”Based on the cost calculation, prepared using the simplified method and prices from the third quarter of 2019 from the Sekocenbud price list and producer prices, the cost of implementing the basic elements of the building's raw state was made according to the three proposed solutions.”, can author provide references for “Price per unit” of each solutions? 4.2 “The analysis of assembly time,” can author discuss the where the figures in Table 6 obtained?

Author Response

Thank you for the good review.

The correct have been marked in yellow (manuscript).

Yours faithfully

Authors

Reviewer 2 Report

There are some weaknesses through the manuscript which need improvement. Therefore, the submitted manuscript cannot be accepted for publication in this form, but it has a chance of acceptance after revise and resubmit. My comments and suggestions are as follows:

1- Abstract gives information on the main feature of the performed study, but some details about obtained results should be added. However, a concise abstract is needed.

2- It is necessary to present aim and main objectives of the study in the last part of introduction.

3- The literature study must be enriched. As the manuscript deals with cost analysis, following paper must be read and cite, as they present cost analysis of technical products:

Procedia Manufacturing, 26:753-762 (2018) Applied Soft Computing, 75:227-232 (2019)

4- The referred standards must be cited in the reference list (e.g., PN-EN 12524)-

5- Equation (2) must be double check! Moreover, original reference of the equation must be mentioned.

6- Many sentences referred directly to the “Authors” and “Authors of this article” which are nor necessary. For instance, in capture of Fig. 3 this refer to the authors should be removed.

7- The main concept in cost analysis is not clear. It should be discuss based on the original reference, assumptions and details.

8- It is suggested to use short (concise) titles in tables. For example, Table 6 does not looks nice.

Author Response

Thank you for the good review.

The correct have been marked in green (manuscript).

Yours faithfully

Authors

Round 2

Reviewer 2 Report

The paper has been revised and the cocerns in the review have been addressed.